# Trends in Beverage Consumption and Related Demographic Factors and Obesity among Korean Children and Adolescents

**DOI:** 10.3390/nu12092651

**Published:** 2020-08-31

**Authors:** Su Bin Hwang, SoHyun Park, Guang-Ri Jin, Jae Hyun Jung, Hyeon Ju Park, Su Hyun Lee, Sangah Shin, Bog-Hieu Lee

**Affiliations:** Department of Food and Nutrition, Chung-Ang University, Gyeonggi-do 17546, Korea; subin_0719@nate.com (S.B.H.); sohyunp612@gmail.com (S.P.); jin_gr@nate.com (G.-R.J.); po48201@nate.com (J.H.J.); h_ju96@naver.com (H.J.P.); suhyun1011@nate.com (S.H.L.)

**Keywords:** beverage consumption, soft drink, obesity, Korea National Health and Nutrition Examination Survey, children and adolescents

## Abstract

It is well known that reducing consumption of sugar is a global public health priority. Beverages were the primary source of total sugar intake from processed foods. However, there are few studies investigating the trend of beverage consumption among children and adolescents in Korea. We examined the overall trend in beverage consumption among 11,996 participants aged 10–18 years who were enrolled in the Korea National Health and Nutrition Examination Survey (KNHANES) (1998–2018). Further, we examined the effect of beverage types on beverage consumption-related demographic factors and obesity among 6121 participants using the recent 24 h dietary recall data (2010–2018) that captured the consumption of fruit and vegetable juices, soft drinks, milk and milk-based products and alcoholic beverages. Demographic characteristics, including sex, age, body mass index, household income level and residential area, were considered. Consumers’ overall beverage intake and the percentage of energy derived from fruit and vegetable juices and soft drinks steadily increased from 1998 to 2016–2018 (*p*-trend < 0.0001); in contrast, dairy product consumption declined since 2010–2012. The main sources of beverage-based calories were fruit and vegetable juices (107.5 kcal/day), soft drinks (145.2 kcal/day), dairy products (181.8 kcal/day) and alcoholic beverages (103.5 kcal/day). Also, Korean adolescents aged 16–18 years consumed more soft drinks, fewer dairy products and higher alcoholic drinks than other age groups; particularly, boys consumed more energy from beverages (*p* < 0.0001). The odds ratios of obesity prevalence tended to be higher for soft drink consumption than for other beverages but this was not significant. The consumption of fruit and vegetable juices and milk and milk products showed a marginal association with a reduced risk of obesity prevalence. Since beverage consumption has increased steadily among Korean children and adolescents, appropriate interventions are needed. In the future, data from a larger sample of Korean children and adolescents are necessary to identify significant differences and longitudinal studies are necessary to examine the causalities.

## 1. Introduction

Rapid economic development has changed the lifestyles of Asian populations. Changes to diet, including an increase in beverage consumption, have contributed to the prevalence of chronic diseases such as obesity, type 2 diabetes and hypertension [1,2]. Consumption of sugar-sweetened beverages (SSBs) increases total caloric intake and is linked to chronic disease risk, including adverse cardiometabolic outcomes [3]. As SSB consumption is prevalent worldwide, in particular, among young adults and men [4,5], population weight gain is likely to continue, driving the global burden of chronic disease [6].

In Korea, from 2008 to 2018, daily beverage consumption more than doubled from an average of 86.6 g to 184.5 g among people older than 1 year [7]; during the same period, beverage sales increased from 11.1% to 14.5% [8]. These data show that the proportion of overall caloric intake accounted for by beverage consumption has been increasing in Korea. The Ministry of Food and Drug Safety has reported that beverages were the primary source of total sugar intake from processed foods in 2013 (31.3%). Meanwhile, sugar-sweetened carbonated beverage (not including club soda) was the most popular beverage type among people aged 6–29 years [9]. There is a positive correlation between beverage intake and obesity rates among Korean children; obesity rates increased steadily during 2001–2015 among Korean children and adolescents [10].

During adolescence, the human body undergoes rapid physiological, psychosocial and emotional development [11]. Dietary habits acquired during childhood tend to consolidate into adulthood [10]. Understanding trends in diet is crucial to developing interventions and policies aimed at preventing diet-related disease. Thus, rigorous research on the impact of dietary factors on health outcomes is required. 

As eating habits and beverage consumption patterns become westernized in Korea, it is expected that the association between beverage consumption and sociodemographic factors and obesity will be affected by beverage types. Therefore, this study examined changes to beverage consumption patterns and associated factors among Korean children and adolescents, enrolled in the Korea National Health and Nutrition Examination Survey (KNHANES). This study also examined the general correlates of consumption per beverage type. 

## 2. Materials and Methods 

### 2.1. Data Source and Participants

KNHANES is a nationwide cross-sectional survey conducted every year and its target population consists of nationally representative non-institutionalized civilians in Korea. Each survey year recruits a new sample of about 10,000 individuals aged 1 year and above. The sampling plan follows a multi-stage clustered probability design. All statistics of this survey have been calculated using sample weights assigned to sample participants that were constructed to represent the Korean population by accounting for the complex survey design, survey non-responses and post-stratification. The weights based on the inverse of selection probabilities and the inverse of response rates were modified by adjusting them to the sex- and age-specific Korean populations (post-stratification). Detailed information on KNHANES is available (http://knhanes.cdc.go.kr/).

This longitudinal study involved 11,996 children and adolescents aged 10–18 years, enrolled in the KNHANES during 1998–2018 to estimate the time trends of beverage consumption and associated energy intake. Further, the cross-sectional study was performed to find out the association between beverage consumption and sociodemographic factors and obesity during 2010–2018. In the cross-sectional study, of 7566 eligible participants, we excluded 1445 participants due to missing information on body mass index (BMI) or body weight or outlier values of energy intake (<500 or >5000 kcal/day). A total of 6,121 children and adolescents were included the final analysis.

The present study complied with the principles of the Declaration of Helsinki; the study protocols for KNHANES I-VII were approved by the Institutional Review Board of Korea Centers for Disease Control and Prevention (KCDC) (IRB no. 2007-02CON-04-P, 2008-04EXP-01-C, 2009-01CON-03-2C, 2010-02CON-21-C, 2011-02CON-06-C, 2012-01EXP-01-2C, 2013-07CON-03-4C, 2013-12EXP-03-5C, 2015-01-02-6C, 2018-01-03-P-A). Informed consent was obtained from the parents or guardians of the children and adolescents prior to participation.

### 2.2. Beverage Consumption and Categories

The KNHANES collects dietary intake data using a single 24 h recall methodology; the type and quantity of foods consumed by participants in the preceding 24 h are recorded during face-to-face interviews. Energy and nutrient intake were estimated using the KNHANES nutrients database based on the food composition table of the Korean rural development administration [2]. In categorizing beverage types, we adapted the categorization reported in previous studies that involved Korean children and adolescents, as follows—fruit and vegetable juices, soft drinks, milk and milk-based products and alcoholic beverages (beer, soju and others) [10,12]. We calculated the total beverage intake (g/day) and total associated energy intake (kcal/day), including as a proportion of the overall energy intake (% energy of total intake); these estimates were calculated per beverage type. We did not assess the water consumption in this study because KNHANES did not collect data on water consumption. In addition, we calculated the intake in only those consumers who consumed a specific beverage type. For example, the consumer of fruit and vegetable juices was defined as a person who only consumed fruit and vegetable juices, excluding other beverages. 

### 2.3. Exposure Variables and Covariates

General demographic characteristics considered were sex, age, body mass index (BMI), household income level and residential area (rural or urban) with parents or guardian’s assistance. Height (Seca 225; SECA, Hamburg, Germany) and weight (GL–6000–20; CASKOREA Co., Ltd., Seoul, Korea) were obtained using standardized techniques and calibrated equipment and with participants wearing light clothes without shoes. BMI was calculated as weight divided by height squared (kg/m^2^); obesity was defined using age- and sex-specific BMI cutoffs proposed by the 2017 Growth Chart from the KCDC [13]. BMI < 5th percentile was classified as underweight; BMI > 5th percentile and < 85th percentile was classified as normal; BMI > 85th percentile and < 95th percentile was classified as overweight; BMI > 95th percentile was classified as obese.

### 2.4. Statistical Analysis

All statistical analyses used sampling weight by the Statistical Analysis System (SAS) survey procedure to consider sampling design of the national survey. The demographic characteristics of consumers and energy intake associated with each beverage type are presented as means ± standard error for continuous variables and as percentage ± standard error of percentage for categorical variables. The chi-square test was used to compare categorical variables; survey regression models were used to analyze linear trends of continuous variables. Associations between beverage consumption and demographic characteristics were examined with two survey logistic regression models performed using (1) “none” versus “any consumption” and “none and < median” versus “≥ median” as outcomes for each beverage type, respectively. Additionally, the association between consumption of beverage type (none, < median and ≥ median) and prevalence of obesity was estimated using the survey logistic regression model. All multivariate analyses were adjusted for age (continuous), sex (boys and girls), BMI (low, normal, overweight and obese), household income level (low, medium-low, medium-high and high), residential area (rural and urban) and energy intake (continuous). All statistical analyses were conducted using SAS 9.4 ver. (SAS Institute Inc., Cary, NC, USA). All *p*-values < 0.05 were considered statistically significant.

## 3. Results

There were 11,996 children and adolescents with complete dietary recall data recorded in the KNHANES for the period between 1998 and 2018. The prevalence of fruit and vegetable juices (11.6–48.7 g/day; *p* for trend < 0.0001) and soft drinks (38.2–102.7 g/day; *p* for trend < 0.0001) consumption among children and adolescents increased significantly across survey cycles. However, while consumption of dairy products increased significantly (130.6–156.4 g/day; *p* for trend < 0.0001) from 1998 to 2012, it declined significantly (156.4–138.4 g/day; *p* for trend < 0.0001) during 2010–2012. No changes to alcoholic beverages consumption were observed (Figure 1A,B).

There were 6,121 children and adolescents (3228 boys and 2893 girls) with complete dietary recall data recorded in the KNHANES for the period between 2010 and 2018.

Sample characteristics, including participants’ sex, age and area of residence were similar across three rounds of surveys (KNHANES V, VI and VII). Meanwhile, average height, weight and BMI significantly increased from 2010–2012 V to 2016–2018. Moreover, the proportion of households with high income significantly increased from 2010 to 2018. Approximately 85% of participants lived in urban areas. The mean energy intake and % energy of total intake declined from 2010–2012 to 2016–2018. However, fat intake and % energy of total intake derived from fats significantly increased across three surveys (Table 1).

The beverage consumption was presented in Table 2. The prevalence of beverage consumption was high for soft drinks (29%) and milk and milk products (51.4%). The total amount intake of total participants and only consumers showed fruit and vegetable juices (43.6 and 248.5 g/day), soft drinks (96.8 and 333.9 g/day), milk and milk products (145.2 and 282.4 g/day) and alcoholic beverages (7.0 and 136.5 g/day). The beverage calories of total and only consumers were fruit and vegetable juices (18.9 and 107.5 g/day), soft drinks (42.0 and 145.2 g/kcal), milk and milk products (93.5 and 181.8 g/kcal) and alcoholic beverages (5.3 and 103.5 g/kcal). Contribution to total energy intake of total and only consumers showed fruit and vegetable juices (0.9 and 5.1% energy of total intake), soft drinks (1.9 and 6.6% energy of total intake), milk and milk products (4.5 and 8.7% energy of total intake) and alcoholic drinks (0.2 and 3.8% energy of total intake).

The consumption of fruit and vegetable juices, soft drinks and alcoholic beverages was significantly higher among participants aged 16–18 years than among any other age group (*p* < 0.0001). Meanwhile, dairy consumption declined with age (*p* < 0.0001). A high BMI (obese) was associated with the consumption of alcoholic beverages (*p* < 0.0001). A higher household income was associated with the consumption of fruit and vegetable juices and dairy products (*p* < 0.0001). A lower household income was associated with the consumption of soft drinks (*p* < 0.0001). Participants resident to urban areas were more likely to consume fruit and vegetable juices, dairy products and alcoholic beverages than rural areas, whereas those resident to rural areas were more likely to consume soft drinks than urban areas (*p* < 0.0001) (Table 3).

The associations between beverage consumption (none versus any consumption) and demographic factors are shown in Table 4. Girls were more likely than boys to consume fruit and vegetable juices (odds ratio [OR]: 1.19; 95% confidence interval [CI]: 1.01–1.42). We identified that age affected the beverage intake pattern and compared with children aged 10–12 years (reference group), those aged 16–18 years were more likely to consume soft drinks (OR: 1.45; 95% CI: 1.23–1.71). Dairy consumption was strongly associated with age; whereby, compared with children aged 10–12 years (reference group), those aged 13–15 years (OR: 0.61; 95% CI: 0.53–0.71) and 16–18 years (OR: 0.42; 95% CI: 0.36–0.49) were less likely to consume dairy products. Low BMI group was more likely to consume dairy products (OR: 1.39; 95% CI: 1.00–1.93) and alcoholic beverages (OR: 1.70; 95% CI: 1.00–2.90) than normal BMI group (reference). Finally, high household income level group was more likely to consume fruit and vegetable juices (OR: 1.59; 95% CI: 1.16–2.18) and less likely to consume soft drinks (OR: 0.69; 95% CI: 0.54–0.89) than low household income level group. Associations between beverage consumption and demographic factors were similar for “none and <median versus ≥ median” and “none versus any consumption” groups. However, girls were less likely than boys to consume fruit and vegetable juices (OR: 0.79; 95% CI: 0.63–0.99). In addition, fruit and vegetable juice consumption (≥median) was associated with age; 13–15 years group (OR: 1.36; 95% CI: 1.07–1.72) and 16–18 years group (OR: 1.41; 95% CI: 1.08–1.84) were more likely to consume fruit and vegetable juices than the reference group (10–12 years group). Moreover, soft drink consumption (≥median) had a strong association with age; 13–15 years group (OR: 1.36; 95% CI: 1.10–1.67) and 16–18 years group (OR: 1.95; 95% CI: 1.58–2.41) were more likely to consume soft drinks than the reference group. Medium-high (OR: 1.32; 95% CI: 1.03–1.70) and high household income level groups (OR: 1.31; 95% CI: 1.02–1.68) were more likely to consume dairy products (≥median) than low and medium-low groups (Table 4).

The fruit and vegetable juices consumption (≥median) showed the marginal association with a reduced risk of obesity prevalence (OR: 0.68; 95% CI: 0.44–1.06) compared to non-drinker. However, the association was attenuated in model 2 adjusted for potential confounders (age, sex, BMI, household income level, residential area and energy intake) (OR: 0.72; 95% CI: 0.46–1.14). The milk and milk products consumption (≥median) tended to be associated with lower risk of prevalence for obesity (OR: 0.73; 95% CI: 0.55–0.97) but association was attenuated after additional adjustment (age, sex, BMI, household income level, residential area and energy intake) (OR: 0.78; 95% CI: 0.58–1.04). Compared to non-drinker, the consumption of soft drinks showed the marginal association with higher risk of becoming obesity but not significant (Table 5).

## 4. Discussion

Our analysis showed that the beverage consumption steadily increased between 1998 and 2018. Boys were more likely to consume beverages than girls. Adolescents aged 16–18 years were more likely to consume fruit and vegetable juices, soft drinks and alcoholic beverages and less likely to consume milk and milk products than the reference group. Consumption of fruit and vegetable juices and milk and milk products tended to be associated with reduced risk of obesity, whereas, soft drinks had the reverse association but not significant.

Previous studies are similar to our beverage consumption trends in Korea from 1998 to 2007–2009 [1] and in particular in developing countries, such as India [14] and Mexico [15], as they experience rapid socioeconomic progress. However, sugar-sweetened beverages (SSBs) consumption have declined recently in the United States from 2003 to 2016, energy and sugar intake from all beverages and SSBs decreased among the total population, children and adults [16]. One study reported that without deliberate policy actions, SSBs are likely to become more affordable and more widely consumed around the world [17]. Therefore, appropriate interventions are needed. 

Compared to solid foods of the same caloric value, liquids are associated with lower satiety and quicker energy conversion [18]. As a result, calories from beverages can easily lead to overconsumption and weight gain [19]. Soft drinks contain excess levels of sugars, caffeine and chemicals such as phosphorus, which interfere with calcium absorption and low mouth pH, leading to tooth decay among children and adolescents [20,21]. The World Health Organization recommends that to prevent chronic diseases, free sugar content should be < 10% of the total caloric intake of a food [22]; the consumption of one or two servings of commercially available beverages already exceeds the recommended daily sugar intake (sugar content/a single serving; carbonated beverage (19.9 g/200 mL); candy (7.1 g/10 g); snack (3.5 g/30 g) [5,23]. In fact, according to the MFDS [24], the consumption of sugar from processed foods among the Korean population aged 3–29 years greatly exceeds the recommended sugar intake by > 10%, this excess consumption is driven by beverages (13.9 g, 31.1%), specifically soft drinks. According to the American Heart Association’s scientific statement in 2017, BMI was significantly correlated with the consumption of SSBs; as the consumption of SSBs increased, rates of overweight and obesity increased as well [25]. In addition, Korean children and adolescents who consume > 10% of their daily calories from processed foods were 39% more likely to develop obesity and 66% more likely to develop hypertension. Due to the westernization of eating habits in Korea, obesity prevalence has increased, the socioeconomic burden of obesity is estimated at 5.6 trillion dollars per year and it continues to increase, highlighting the importance of curtailing excess sugar consumption [26].

In this study, no significant association was found between soft drinks consumption and change in BMI. The consumption of soft drinks showed marginal association with increased risk of obesity prevalence, whereas, the consumption of fruit and vegetable juices and milk and milk products showed marginal association with reduced risk of obesity prevalence. Fruit and vegetable juices contain a variety of polyphenols and vitamins. Drinking juices is an efficient way to improve consumption of fruit and vegetables. Previous studies showed similar results to ours that phytochemicals from fruit and vegetable juices reduce the risk of obesity-induced chronic disease by improving the metabolic profiles such as lowering blood pressure, improving blood lipid profiles and reducing systemic inflammation [27,28]. Some juices, for example, mulberry and blueberry juice with rich amounts of anthocyanin reduced body weight, downregulated serum cholesterol, insulin resistance, lipid accumulation and leptin secretin in high-fat-diet induced obese mice [29]. Also, the polyphenols in plum and peach juice have shown the weight-reducing effects [30].

Moreover, Beck (2017) has shown that higher milk fat consumption is associated with lower correlation with severe obesity among Latino preschoolers [31]. In addition, a number of cross-sectional studies have reported an inverse correlation between BMI or adiposity and dairy product consumption in children, which is lined with our results. Dairy products contain several components including proteins, vitamins D, calcium, phosphorus and fat that may contribute to lower risk of becoming overweight or obese. Whey protein from milk stimulates insulin secretion that may directly affect food intake regulation by suppressing appetite [32]. Calcium has been suggested to play a key role in energy metabolism by forming insoluble soaps or binding bile acids. Moreover, calcium may play a key role in intracellular pathways both directly and indirectly through 1.25-dihydroxy vitamin D and calcitriols [33]. Milk fat is one of the few food sources of butyric acid which has anti-inflammatory effects, improves insulin sensitivity and increase energy expenditure in mice models. Furthermore, full fat milk may have increased satiety for children in comparison with lower fat options. Thus, a possibility is that these properties of dairy components are protective against obesity and metabolic dysfunction [31]. Encouraging milk and dairy consumption is therefore important in protecting against obesity. 

Dietary recommendations for adolescents encourage daily consumption of > 2 cups of plain white milk (not including processed milk) [34]; the presented survey findings suggest that milk intake among Korean children and adolescents is insufficient. A survey on milk intake among middle and high school students in Korea revealed that 73.2% of participants drink milk less than 2 cups, which is insufficient for recommendation consumption [35]. KNHANES VII has indicated that the mean daily intake of calcium for boys and girls aged 10–18 years was 594.4 mg and 511 mg, respectively, which was below the recommended doses of 900 mg and 833 mg, respectively [7,36]. Low calcium intake in children and adolescents aged 10–18 years may be associated with low dairy intake, which is the main source of calcium for this age group [37]. 

A study by Lee (2019) has shown that participating in the school milk program directly resulted in higher calcium intake and improved the overall nutrient intake [38]. Milk consumption improves dietary calcium and overall nutrient intake in adolescents. In a study of elementary school students in Seoul, overall nutrient intake, including that of calcium and vitamin B_2_, was higher among participants who consumed more milk and dairy products than among those who did not [38]. Although the present findings suggest that dairy consumption has been declining, they also indicate between-age group differences in consumption, which should be addressed. Availability of milk at schools appears to positively affect the intake of calcium and other important nutrients among Korean children and adolescents [39], suggesting milk should be available at schools and its consumption should be mandatory to improve young people’s nutrition.

Furthermore, we investigated the association of beverage consumption related to sociodemographic factors. A study in the United States revealed that lower household income was associated with higher consumption of sugary drinks [40]. Furthermore, Shim (2009) found significant differences in milk intake between primary and middle school students according to their household income, whereby low household income was associated with low milk consumption [41]. The present study findings are consistent with these findings. The dietary habits of adolescents are influenced by sociodemographic factors. Overall, low levels of education and income are associated with the low likelihood of maintaining a healthy diet. In contrast, higher socioeconomic status is associated with healthier meals [11]. Adolescents living in high-income households are more likely to be exposed to dietary information and develop an interest in healthy foods such as vegetables and fruits [42]. Nutrition education programs should account for participants’ sociodemographic background, as it affects individuals’ dietary habits [42].

In the present study, the boys had a higher beverage intake than girls for all beverage types. Assuming that boys generally have bigger bodies and higher muscle mass and metabolic rates than girls, boys need and consume more energy than girls. Hence, it is possible that boys need more beverage consumption than girls. Compared with other age groups, participants aged 16–18 years were most likely to consume alcoholic beverages. Alcohol consumption in early adulthood has been associated with the increased risk of accidents, violence, suicide and poor academic performance and failure; excessive drinking can cause brain damage and lead to long-term problems [43]. Drinking habits developed in adolescence are difficult to change during adulthood [44]. According to the KCDC, the annual rate of participation in drinking prevention education was low at 38.8% in 2016, 41.2% in 2017 and 42% in 2018 among Korean children and adolescents. In addition, the nutrition education programs in school was low at 45.3% in 2016, 46.2% in 2017 and 47.2% in 2018 among Korean children and adolescents. Educational programs focused on nutrition and dietary habits should incorporate drinking prevention elements [45,46,47].

This study has several limitations, which should be considered when interpreting its findings. First, this is the first study on beverage consumption among Korean children and adolescents aged 10–18 years, which precludes direct comparisons to previous studies. Second, as beverage consumption was assessed using a single 24 h recall, exclusive reliance on self-report data could introduce bias, such as memory bias. Furthermore, the sociodemographic characteristics may have biased our findings because of underreporting, unreliability and conversion errors [48]. Third, this study did not comprehensively examine all drink types consumed in Korea. Fourth, our cross-sectional study was limited in determine the causalities. Finally, even though we adjusted for some confounding factors, residual confounders may remain constrained by unavailable variables, such as an unbalanced food pattern, lacking in micro- and macro nutrients and physical activity.

These limitations notwithstanding, the present study has several strengths, including a large nationally representative sample of the Korean population. In addition, we included data from children and adolescents, which were collected over a decade (1998–2018). Most previous studies set in Korea involved adult participants. Although some previous studies examined beverage consumption, they tended to focus on a few specific beverages. In contrast, our study identified trends in consumption of beverage types commonly consumed by Korean children and adolescents aged 10–18 years.

## 5. Conclusions

Beverage consumption among Korean children and adolescents has significantly increased during 1998–2018. The absolute amounts of all beverage types intakes were higher among boys than among girls. Among all age groups, the consumption of dairy products was the highest among children aged 10–12 years; however, it was below the recommended levels. The intake of soft drinks and alcoholic beverages was significantly higher among adolescents aged 16–18 years than among other age groups. Compared with low household income, high income was associated with higher consumption of fruit and vegetable juices and dairy products and lower consumption of soft drinks. The ORs of obesity prevalence tended to be lower in fruit and vegetable juices and milk and milk products. However, marginal increased risk of obesity was observed in soft drinks but not significant. The present findings provide foundational evidence for formulating intervention strategies and policies that promote healthy eating habits among children and adolescents and target high-risk populations in South Korea. In addition, future longitudinal studies are needed to test direct causality between beverage consumption and obesity and data from a larger sample of Korean children and adolescents is required to identify significant differences.

## Figures and Tables

**Figure 1 nutrients-12-02651-f001:**
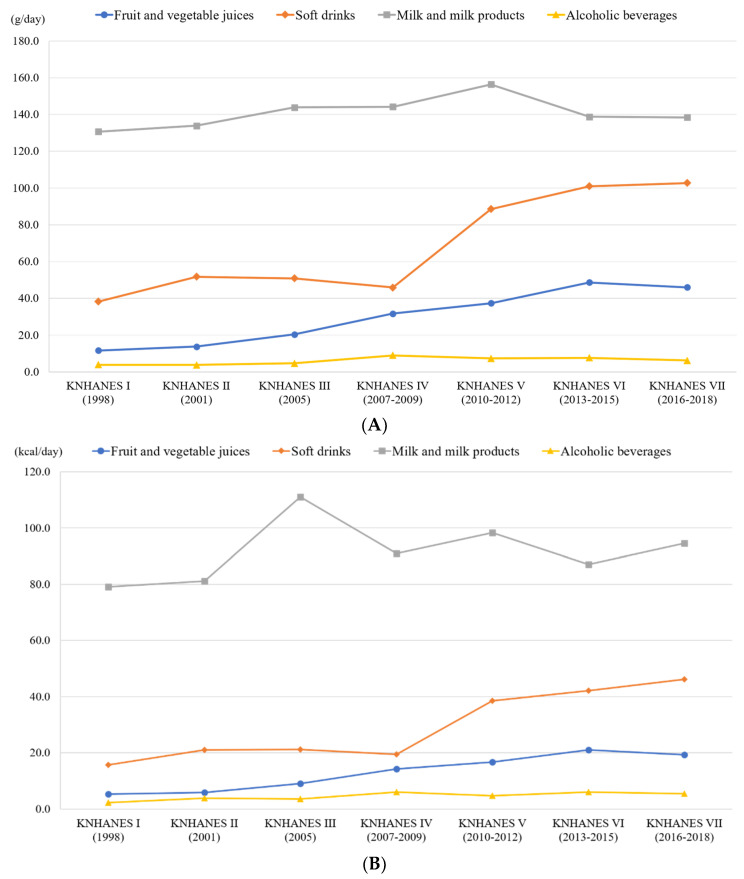
Trends in beverage consumption by beverage types from 1998 to 2018 among Korean children and adolescents. (**A**) Trends in consumption (g/day) by beverage types from 1998 to 2018 among Korean children and adolescents. (**B**) Trends in calorie intake (kcal/day) by beverage types from 1998 to 2018 among Korean children and adolescents.

**Table 1 nutrients-12-02651-t001:** Sample size by survey year and demographic characteristics of subjects who completed Korea National Health and Nutrition Examination Survey (KNHANES) 2010–2012, 2013–2015 and 2016–2018.

	KNHANES V (2010–2012) (*N* = 2394)	KNHANES VI (2013–2015) (*N* = 1996)	KNHANES VII (2016–2018) (*N* = 1731)	*p*-Value ^(2)^
Sex				0.6728
Boys	1259 (53.3) ^(1)^	1065 (52.0)	904 (52.1)	
Girls	1135 (46.7)	931 (48.0)	827 (47.9)	
Age				0.6422
10–12 years	980 (33.8)	750 (32.7)	675 (32.4)	
13–15 years	829 (33.7)	684 (33.4)	576 (32.4)	
16–18 years	585 (32.5)	562 (33.9)	480 (35.2)	
Anthropometry				
Height (cm)	160.6 ± 0.3	160.6 ± 0.3	161.3 ± 0.3	0.0162
Weight (kg)	53.4 ± 0.4	54.5 ± 0.4	54.8 ± 0.4	<0.0001
BMI (kg/m^2^)	20.4 ± 0.1	20.9 ± 0.1	20.8 ± 0.1	<0.0001
Waist circumference (cm)	68.8 ± 0.3	70.2 ± 0.3	70.0 ± 0.3	0.236
Household income level				0.0213
Low	264 (14.1)	217 (11.6)	170 (10.4)	
Medium-low	595 (29.0)	522 (27.6)	404 (24.0)	
Medium-high	757 (29.7)	665 (33.0)	573 (33.1)	
High	750 (27.2)	578 (27.9)	582 (32.4)	
Residential area				0.1194
Rural	343 (17.6)	327 (16.7)	224 (12.0)	
Urban	2051 (82.4)	1669 (83.3)	1507 (88.0)	
Nutrient intake				
Energy (kcal/day)	2147.7 ± 20.7	2138.1 ± 21	2106.9 ± 23.9	<0.0001
Carbohydrate (g/day)	332.6 ± 3.2	320.1 ± 3.2	313.6 ± 3.9	0.3723
% energy of total intake	63.0	60.6	60.3	<0.0001
Protein (g/day)	77.5 ± 1.0	76 ± 1.0	77.2 ± 1.1	<0.0001
% energy of total intake	14.3	14.1	14.6	0.7072
Fat (g/day)	56.4 ± 0.9	58.1 ± 0.9	57.8 ± 1.0	<0.0001
% energy of total intake	22.8	23.9	24.1	<0.0001

^(1)^ Values are presented as N (%) and means ± standard error. ^(2)^
*p*-values were calculated by general linear regression for continuous variables and chi-square for categorical variables.

**Table 2 nutrients-12-02651-t002:** Beverage consumption of the Korean children and adolescents aged 10–18 years in the KNHANES V, VI and VII (2010–2018).

Beverage Category	Intake of Only Subjects (*n* (%))	Total Amount Intake (g/Day)		Total Amount Energy Intake (kcal/Day)		Contribution to Total Energy Intake (% Energy)	
	Total (*n* = 6121)	Total Subjects	Consumer Only	Total Subjects	Consumer Only	Total Subjects	Consumer Only
Fruit and vegetable juices	1057 (17.6) ^(1)^	43.6 ± 2.0 ^(2)^	248.5 ± 7.1	18.9 ± 0.9	107.5 ± 3.3	0.9	5.1
Fruit juices	862 (14.3)	38.4 ± 1.9	269.0 ± 7.7	17.0 ± 0.9	119.2 ± 3.6	0.8	5.7
Vegetable juices	238 (4.0)	5.2 ± 0.7	129.3 ± 8.2	1.9 ± 0.2	46.2 ± 4.0	0.1	2.3
Soft drinks	1700 (29.0)	96.8 ± 3.7	333.9 ± 9.0	42.0 ± 1.7	145.2 ± 4.1	1.9	6.6
Milk and milk products	3310 (51.4)	145.2 ± 3.1	282.4 ± 4.2	93.5 ± 2.0	181.8 ± 2.7	4.5	8.7
Milk	2730 (41.8)	123.6 ± 3.0	295.8 ± 4.5	75.8 ± 1.8	181.4 ± 2.8	3.6	8.6
Milk products	1096 (17.3)	21.6 ± 1.0	125.2 ± 4.0	17.7 ± 0.9	102.6 ± 3.5	0.9	5.1
Alcoholic beverages	316 (5.2)	7.0 ± 1.6	136.5 ± 29.1	5.3 ± 1.2	103.5 ± 19.4	0.2	3.8
Beer	34 (0.7)	4.4 ± 1.3	593.8 ± 125.5	1.7 ± 0.5	233.1 ± 48.1	0.1	7.8
Soju	154 (2.6)	2.1 ± 0.6	78.0 ± 17.1	3.2 ± 0.9	122.2 ± 24.9	0.1	4.6
Others	153 (2.3)	0.6 ± 0.3	27.2 ± 11.2	0.4 ± 0.2	17.6 ± 6.3	0.0	0.7

^(1)^ Values are presented as N (%). ^(2)^ Values are presented as means ± standard error.

**Table 3 nutrients-12-02651-t003:** Mean (± standard error) beverage consumption (in g/day and kcal/day) of the Korean children and adolescents by general characteristic in the KNHANES V, VI and VII (2010–2018).

Beverage Category	Fruit and Vegetable Juices		Soft Drinks		Milk and Milk Products		Alcoholic Beverages	
	(g/Day)	(kcal/Day)	(g/Day)	(kcal/Day)	(g/Day)	(kcal/Day)	(g/Day)	(kcal/Day)
Sex								
Boys	46.7 ± 3.0 ^(1)^	20.2 ± 1.3	117.8 ± 5.7	51.0 ± 2.6	165.2 ± 4.5	104.9 ± 2.9	9.6 ± 2.8	6.8 ± 1.7
Girls	40.2 ± 2.6	17.4 ± 1.2	73.3 ± 4.1	31.8 ± 1.8	123.0 ± 4.1	80.8 ± 2.7	4.2 ± 1.4	3.7 ± 1.5
Age								
10–12 years	35.5 ± 2.4	15.4 ± 1.0	64.9 ± 3.8	27.9 ± 1.7	178.7 ± 5.0	114.2 ± 3.2	0.1 ± 0.0	0.2 ± 0.0
13–15 years	45.1 ± 3.4	19.9 ± 1.5	93.1 ± 6.0	41.1 ± 3.0	149.4 ± 5.3	95.1 ± 3.4	0.8 ± 0.5	0.4 ± 0.2
16–18 years	49.4 ± 4.1	20.9 ± 1.8	128.3 ± 7.4	55.1 ± 3.2	111.8 ± 5.2	73.8 ± 3.5	19.1 ± 4.6	14.7 ± 3.3
BMI								
Low	44.1 ± 9.7	19.1 ± 4.5	99.3 ± 17.4	43.1 ± 7.3	138.7 ± 16.5	89.8 ± 10.1	3.2 ± 2.1	3.7 ± 3.0
Normal	43.5 ± 2.2	18.8 ± 1.0	97.8 ± 4.4	42.5 ± 2.0	147.5 ± 3.5	95.2 ± 2.3	7.0 ± 1.9	5.2 ± 1.4
Overweight	46.0 ± 5.9	20.5 ± 2.8	84.6 ± 8.6	36.0 ± 3.7	148.5 ± 8.2	92.4 ± 5	5.7 ± 2.6	6.4 ± 3.2
Obese	40.9 ± 7.3	17.5 ± 3.2	102.1 ± 10.2	44 ± 4.5	122.7 ± 10.4	81.1 ± 6.9	10.8 ± 7.7	6.6 ± 4.2
Household income level								
Low	37.5 ± 6.0	16.2 ± 2.7	112.6 ± 10.1	48.1 ± 4.3	125.0 ± 8.6	82.4 ± 5.7	10.8 ± 8.1	5.0 ± 3.2
Medium-low	40.9 ± 3.9	17.7 ± 1.7	105.6 ± 7.3	47.3 ± 3.7	126.2 ± 5.6	81.5 ± 3.7	5.6 ± 2.5	5.1 ± 2.5
Medium-high	37.2 ± 3.0	16.5 ± 1.3	97.3 ± 7.3	41.4 ± 3.0	155.3 ± 5.6	99.5 ± 3.6	4.5 ± 1.2	4.8 ± 1.6
High	55.8 ± 4.3	23.6 ± 1.8	80.9 ± 6.2	34.8 ± 2.7	161.0 ± 5.6	103.2 ± 3.7	9.8 ± 3.8	6.5 ± 2.5
Residential area								
Rural	36.1 ± 4.4	15.1 ± 1.9	101.3 ± 8.5	44.3 ± 3.7	140.3 ± 8.0	91.3 ± 5.2	6.5 ± 5.9	2.8 ± 2.2
Urban	45.0 ± 2.2	19.6 ± 1.0	95.9 ± 4.1	41.5 ± 1.9	146.2 ± 3.4	93.9 ± 2.2	7.2 ± 1.6	5.8 ± 1.3

^(1)^ Values are presented as means ± standard error. All *p*-values were < 0.0001.

**Table 4 nutrients-12-02651-t004:** Association between general characteristics and types of beverage consumption among the Korean children and adolescents based on the KNHANES V, VI and VII (2010–2018).

Beverage Category	Fruit and Vegetable Juices				Soft Drinks				Milk and Milk Products				Alcoholic Beverages			
	None Vs. any Consumption		None and < Median Vs. ≥ Median		None Vs. any Consumption		None and < Median Vs. ≥ Median		None Vs. any Consumption		None and < Median Vs. ≥ Median		None Vs. any Consumption		None and < Median Vs. ≥ Median	
	OR	95% CI	OR	95% CI	OR	95% CI	OR	95% CI	OR	95% CI	OR	95% CI	OR	95% CI	OR	95% CI
Sex																
Boys	1.00		1.00													
Girls	1.19	(1.01–1.42)	0.79	(0.63–0.99)	0.96	(0.83–1.10)	0.91	(0.76–1.10)	1.03	(0.90–1.17)	0.83	(0.71–0.96)	1.07	(0.82–1.41)	0.89	(0.60–1.31)
Age																
10–12 years	1.00		1.00													
13–15 years	1.04	(0.87–1.24)	1.36	(1.07–1.72)	1.08	(0.92–1.27)	1.36	(1.10–1.67)	0.61	(0.53–0.71)	0.72	(0.61–0.84)	0.74	(0.52–1.05)	0.82	(0.52–1.31)
16–18 years	1.09	(0.90–1.33)	1.41	(1.08–1.84)	1.45	(1.23–1.71)	1.95	(1.58–2.41)	0.42	(0.36–0.49)	0.52	(0.42–0.63)	1.90	(1.40–2.59)	2.60	(1.67–4.05)
BMI																
Low	1.28	(0.83–2.00)	0.87	(0.50–1.52)	0.89	(0.63–1.26)	0.81	(0.52–1.27)	1.39	(1.00–1.93)	1.30	(0.89–1.90)	1.70	(1.00–2.90)	1.15	(0.52–2.55)
Normal	1.00		1.00								0.87	(0.70–1.09)			0.82	(0.45–1.50)
Overweight	1.05	(0.83–1.33)	1.03	(0.74–1.44)	0.86	(0.70–1.06)	0.80	(0.61–1.05)	0.99	(0.82–1.19)	0.77	(0.58–1.02)	0.86	(0.58–1.30)	0.59	(0.27–1.28)
Obese	0.96	(0.71–1.29)	0.69	(0.44–1.07)	1.11	(0.87–1.43)	1.03	(0.77–1.37)	0.95	(0.75–1.20)			0.74	(0.44–1.24)		
Household income level																
Low	1.00		1.00													
Medium-low	1.22	(0.88–1.68)	1.15	(0.77–1.74)	0.81	(0.63–1.05)	0.90	(0.66–1.24)	0.85	(0.68–1.06)	1.06	(0.82–1.37)	0.99	(0.57–1.72)	0.71	(0.32–1.58)
Medium-high	1.30	(0.96–1.77)	1.03	(0.70–1.51)	0.75	(0.58–0.95)	0.75	(0.55–1.02)	1.17	(0.94–1.46)	1.32	(1.03–1.70)	1.21	(0.73–2.01)	1.21	(0.60–2.46)
High	1.59	(1.16–2.18)	1.48	(1.02–2.16)	0.69	(0.54–0.89)	0.65	(0.47–0.89)	1.19	(0.95–1.47)	1.31	(1.02–1.68)	1.37	(0.84–2.25)	1.25	(0.62–2.52)
Residential area																
Rural	1.00		1.00													
Urban	1.17	(0.90–1.51)	1.24	(0.91–1.70)	0.93	(0.76–1.13)	0.94	(0.73–1.21)	0.92	(0.77–1.11)	0.93	(0.75–1.15)	1.41	(0.92–2.14)	1.55	(0.82–2.91)

OR = odd ratio. Adjusted: for age (continuous), sex (boys and girls), BMI (low, normal, overweight and obese), household income level (low, medium-low, medium-high and high), residential area (rural and urban) and energy intake (continuous).

**Table 5 nutrients-12-02651-t005:** Multivariate adjusted odds ratios and 95% CI for obesity according to consumption of beverage types.

	Non-Drinker	<Median *		≥Median *		*p* for Trend
	OR	OR	95% CI	OR	95% CI	
Fruit and vegetable juices						
model 1 **	1.00	1.19	(0.83–1.71)	0.68	(0.44–1.06)	0.22
model 2	1.00	1.25	(0.87–1.79)	0.72	(0.46–1.14)	0.38
Soft drinks						
model 1	1.00	1.10	(0.78–1.53)	1.09	(0.82–1.45)	0.60
model 2	1.00	1.14	(0.81–1.60)	1.10	(0.85–1.51)	0.39
Milk and milk products						
model 1	1.00	1.04	(0.79–1.37)	0.73	(0.55–0.97)	0.05
model 2	1.00	1.08	(0.82–1.42)	0.78	(0.58–1.04)	0.15
Alcoholic beverages						
model 1	1.00	0.85	(0.43–1.66)	0.55	(0.25–1.18)	0.17
model 2	1.00	0.87	(0.44–1.72)	0.58	(0.27–1.26)	0.21

* median consumption value of each beverage; fruit and vegetable juices (boy: 208 g, girl: 187.2 g), soft drinks (boy: 280.55 g, girl: 210 g), milk and milk products (boy: 249.6 g, girl: 212 g) and alcoholic beverages (boy: 1.21 g, girl: 0.76 g). ** model 1 adjusted for age (continuous), sex (boys and girls); model 2 adjusted for age (continuous), sex (boys and girls), BMI (low, normal, overweight and obese), household income level (low, medium-low, medium-high and high), residential area (rural and urban) and energy intake (continuous).

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
