# Peer review of "Trends in Beverage Consumption and Related Demographic Factors and Obesity among Korean Children and Adolescents"

_nutrients, 2020, doi:10.3390/nu12092651_

Round 1

Reviewer 1 Report

The authors have executed and reported an interesting study assessing beverage consumption trends over the time period of 2010-2018 (or 1998-2018?) among Korean children and adolescents. Overall the methods, results, and discussion were difficult to follow and would benefit from a careful readthrough and re-organization.

Abstract

  • It doesn’t appear the abstract is fully representative of the entire manuscript. Are you presenting data in the abstract from the n=11,996 cohort?
  • Consider some points of discussion in your abstract, as well as a future direction

Introduction

  • Lines 42-43: Why might sugar-sweetened beverage consumption be higher in young adults and men?
  • Line 46: You don’t need the words ‘respectively’ (x2)
  • Line 45: are these data among adults?
  • You go back and forth between discussing sugar sweetened beverage intake and beverage intake in general. Beverage intake in general could include increased water intake which would be a positive nutrition direction. Consider being careful about how you are presenting the available data. Also, in the second paragraph you mention carbonated beverages; is it to be assumed that carbonated beverages are sugar-sweetened (i.e., not club soda)?
  • Line 51: Consider describing the correlation between beverage intake and obesity rates (e.g., a positive correlation?)
  • Consider including a hypothesis(ese)

Materials and Methods

  • Overall, the methods are confusing. I am unsure which data sets were used to answer which research questions (e.g., n=6,121 vs. n=11,996). Is this a cross-sectional study or a longitudinal one? Please clarify.
  • Lines 69-73: did the same 6,121 children/adolescents complete all three components (health interview, health examination, nutrition survey) three times between 2010-2018? Is this the data that is presented in Table 2? If so, it is not clear what data is presented here; have the consumption of certain beverages changed overtime within the same cohort of participants?
  • Line 81: Did you obtain participant assent from the children and adolescents, and parent/guardian consent?
  • Line 81: Consider ‘signed the informed consent’ instead of ‘wrote the informed consent’
  • Line 93: How did you obtain household income level information from children and adolescents? Did their parent/guardian answer this question?
  • Lines 94-95: How did you obtain height and weight? Was this with a scale and stadiometer? Or self-report?
  • In the 24h recall, did you collect water consumption as well? Was this included in the total beverage intake?
  • Lines 100-101: Consider expanding your explanation; I am unsure what you mean by using sampling weight to consider sampling design.
  • Lines 102-104: Why did you report standard error (as opposed to standard deviation)?
  • Why not use the categories of none, < median, and > median when assessing the associations between beverage consumption and demographic characteristics?

Results

  • Lines 118-119: Consider re-wording. The way this is written (sample characteristics are similar across three surveys) sounds like you are implying different participants were recruited for the three ‘surveys’ of the health interview, health examination, and nutrition survey.
  • Consider clarifying the % energy is % energy of total intake
  • Lines 155-161: please include the correlation coefficients and p-values
  • Figures 1a and b: Please include a label on the y-axis
  • Line 180 and Line 183: what were the ‘potential confounders’ and ‘additional adjustments’?

Discussion:

  • Overall the discussion was hard to follow; consider re-organizing
  • Lines 192-201: Nice overview of your findings (results); consider putting this in the results section. The discussion should highlight your main findings and provide rationale for those findings
  • Please define/clarify what you mean by “only consumers”
  • Line 197: What is the reference group? Is this the children aged 10-12 mentioned on line 164? Why are they the reference group?
  • Line 202-203: add a reference for the statement: “Compared to solid foods of the same caloric value, liquids are associated with lower satiety and quicker energy conversion”
  • Line 214-215: is this data from Korea? Please add a reference
  • Lines 215-217: is this data from the world? Please add a reference
  • Lines 217-219: was this US data over the same timeframe as your study (2010-2018)?
  • Lines 223-231: do the guidelines include flavored milk in its daily dairy consumption recommendation (i.e., chocolate milk that includes excess sugar in addition to calcium and other important micronutrients)?
  • Lines 246-249: Are these associations from the present study? If so, please clarify this. If not, please include references
  • Lines 259-261: What is the “annual education participation rate of nutrition and eating habits”?
  • Lines 263-271: nice overview of the importance of fruit and vegetable intake. Consider relating this back to your own findings.
  • Lines 288-292: Consider expanding on the limitations of the 24h recall, and including a reference. You mention the study did not comprehensively examine all drink types consumed in Korea. Does this mean some other drink types were reported in the 24h recalls that you did not include? Is water one of them?
  • Height, weight, BMI increased between 2010-12 and 2016-18; what happened between 2012-16?
  • Consider offering an explanation as to why the intake of all beverage types was higher among boys (is this absolute or relative?)
  • Consider offering some future directions

Author Response

We thank the reviewer for their time and expertise in reviewing the manuscript entitled "Trends in beverage consumption and related demographic factors and obesity among Korean children and adolescents."

Revisions were made according to the reviewer’s comments in the text and are highlighted in red color. Detailed explanations about the changes we made are attached.

We are confident that we have adequately addressed the reviewers’ concerns.

We hope you agree with the importance of our findings and now find the manuscript suitable for publication.

Sincerely,

Bog-Hieu Lee

Department of Food and Nutrition, Chung-Ang University, Gyeonggi-do 17546, Korea

Tel.: +82-31-670-3276

lbheelb@cau.ac.kr

Sangah Shin

Department of Food and Nutrition, Chung-Ang University, Gyeonggi-do 17546, Korea

Tel.: +82-31-670-3259

ivory8320@cau.ac.kr

Reviewer 2 Report

This work aimed to investigate the trend of beverage consumption among Korean children and adolescents and to associate it with demographics and BMI.

It is an interesting topic for a retrospective study, which involves a large number of cases, but it is appropriate to point out some facts.

Introduction: The introduction is broad and clear, summarizing the scientific background and the aims of the authors.

Methods: In order to compare the data with other studies, it would be advisable to use the BMI classification proposed by the WHO, which is the most widely used worldwide.

Results:

  • Tables (and corresponding legends) should be better distinguished and separated from the text
  • Text: Cumbersome writing, if data are shown in the tables, the numbers must not be repeated in the text

Discussion: Often the consumption of soft drinks and alcoholic beverages, even in a context of differences between low and high-incomes, are associated with an unbalanced food pattern, lacking in micro- and macro nutrients, and this fact could also be a confounding factor in the results. Include a comment on this point as a limit of the study.

Among the limitations, include the retrospective nature of the study and the fact that the food history is carried out with 24h recall, which is a useful tool but with limited reliability due to individual compliance.

Author Response

(The authors gave the same response as above.)
